# PyPlutchik: Visualising and comparing emotion-annotated corpora

**Alfonso Semeraro** *, **Salvatore Vilella** , **Giancarlo Ruffo**

Department of Computer Science, University of Turin, Turin, Italy

* alfonso.semeraro@unito.it

## Abstract

The increasing availability of textual corpora and data fetched from social networks is fuelling a huge production of works based on the model proposed by psychologist Robert Plutchik, often referred simply as the "Plutchik Wheel". Related researches range from annotation tasks description to emotions detection tools. Visualisation of such emotions is traditionally carried out using the most popular layouts, as bar plots or tables, which are however sub-optimal. The classic representation of the Plutchik's wheel follows the principles of proximity and opposition between pairs of emotions: spatial proximity in this model is also a semantic proximity, as adjacent emotions elicit a complex emotion (a primary dyad) when triggered together; spatial opposition is a semantic opposition as well, as positive emotions are opposite to negative emotions. The most common layouts fail to preserve both features, not to mention the need of visually allowing comparisons between different corpora in a blink of an eye, that is hard with basic design solutions. We introduce PyPlutchik the Pyplutchik package is available as a Github repository (http://github.com/alfonsosemeraro/pyplutchik) or through the installation commands *pip* or *conda*. For any enquiry about usage or installation feel free to contact the corresponding author, a Python module specifically designed for the visualisation of Plutchik's emotions in texts or in corpora. PyPlutchik draws the Plutchik's flower with each emotion petal sized after how much that emotion is detected or annotated in the corpus, also representing three degrees of intensity for each of them. Notably, PyPlutchik allows users to display also primary, secondary, tertiary and opposite dyads in a compact, intuitive way. We substantiate our claim that PyPlutchik outperforms other classic visualisations when displaying Plutchik emotions and we showcase a few examples that display our module's most compelling features.

## 1 Introduction

The recent availability of massive textual corpora has enhanced an extensive research over the emotional dimension underlying human-produced texts. Sentences, conversations, posts, tweets and many other pieces of text can be labelled according to a variety of schemes, that refer to as many psychological theoretical frameworks. Such frameworks are commonly divided into *categorical* models [1–4], based on a finite set of labels, and *dimensional* models

**Data Availability Statement:** All data is freely available and the plots can be easily reproduced, we do not have any special access to the data. The data is properly referenced in the bibliography, the links are the following: SSEC data: http://www.romanklinger.de/ssec/ Amazon data: https://

jmcauley.ucsd.edu/data/amazon/ IMDB data: https://www.kaggle.com/omarhanyy/imdb-top-1000 We added the scripts to reproduce the plots to the Github repository.(https://github.com/alfonsosemeraro/pyplutchik).

**Funding:** The author(s) received no specific funding for this work.

**Competing interests:** The authors have declared that no competing interests exist.

[5, 6], that position data points as continuous values in an N-dimensional vector space of emotions. One of the most famous dimensional models is Russel's *circumplex model* of emotions [7]. Russel's model posits that affect states are defined by dimensions that are not independent: emotions can be represented as points in a space whose main orthogonal axes represent their degree of *arousal* and of *pleasure/displeasure*. According to Russel, the full spectrum of emotions can be meaningfully placed along the resulting circumference. Russel is not the first scholar to use a circular layout to represent emotions (for instance, it was already used in [8]): indeed, this kind of *circumplex* representation became very popular over time, since it is suitable to spatially represent emotions on a continuous space.

The cultural variation of emotions has also been studied. This is a crucial factor to take into account when classifying emotions: intuitively, the categories themselves can radically change depending on the cultural background [9]. This is also valid with respect to how the annotator perceives the emotions in a text and, if the annotated data is then used to train a classifier, it can introduce a bias in the model. The nomenclature of emotions, their meanings and the relations of words to emotion concepts depend on the social frameworks in which we are born and raised: therefore, cultural variation can significantly impact this kind of analysis. In this regard, in [10] the authors estimate emotion semantics across a sample of almost 2500 spoken languages, finding high variability in the meaning of emotion terms, but also evidence of a *universal structure* in the categorization of emotions. Following different methodological approaches, similar results were previously obtained in [11].

Regardless of their categorical or dimensional nature, the emotional models provide a complex and multifaceted characterisation of emotions, which often necessitates dedicated and innovative ways to visualise them. This is the case of Plutchik's model of emotions [12], a categorical model based on 8 labels (*Joy*, *Trust*, *Fear*, *Surprise*, *Sadness*, *Disgust*, *Anger* and *Anticipation*). According to the model, emotions are displayed in a flower-shaped representation, famously known as *Plutchik's wheel*, which has become since then a classic reference in this domain. The model, that displays emotions in this *circumplex*-like representation and that is described in detail in Sec. 2, leverages the disposition of the *petals* around the wheel to highlight the similar (or opposite) flavour of the emotions, as well as how similar emotions, placed in the same "hemisphere" of the wheel, can combine into primary, secondary and tertiary *dyads*, depending on how many petals away they are located on the flower.

It is clear that such a complex and elaborated solution plays a central role in defining the model itself. Still, as detailed in Sec. 2, many studies that resort to Plutchik's model display their results using standard data visualisation layouts, such as bar plots, tables, pie charts and scatter plots, most likely due to the lack of an easy, plug-and-play implementation of the Plutchik's wheel.

On these premises, we argue that the most common layouts fail to preserve the characterising features of Plutchik's model, not to mention the need of visually allowing comparisons between different corpora at a glance, that is hard with basic design solutions. We contribute to fill the gap in the data visualisation tools by introducing *PyPlucthik*, a Python module for visualising texts and corpora annotated according to the Plutchik's model of emotions. Given the preeminence of Python as a programming language in the field of data science and, particularly, in the area of Natural Language Processing (NLP), we believe that the scientific community will benefit from a ready-to-use Python tool to fulfil this particular need. Of course, other packages and libraries may be released for other languages in the future.

PyPlutchik provides an off-the-shelf Python implementation of the Plutchik's wheel. Each petal of the flower is sized after the amount of the correspondent emotion in the corpus: the more traces of an emotion are detected in a corpus, the bigger the petal is drawn. Along with

**Table 1. Plutchik's 8 basic emotions with 3 degrees of intensity each.** Emotions are commonly referred as the middle intensity degree ones.

| Lower intensity | Emotion | Higher intensity |
|---|---|---|
| Annoyance | **Anger** | Rage |
| Interest | **Anticipation** | Vigilance |
| Serenity | **Joy** | Ecstasy |
| Acceptance | **Trust** | Admiration |
| Apprehension | **Fear** | Terror |
| Distraction | **Surprise** | Amazement |
| Pensiveness | **Sadness** | Grief |
| Boredom | **Disgust** | Loathing |

the 8 basic emotions, PyPlutchik displays also three degrees of intensity for each emotion (see Table 1).

PyPlutchik is built on top of Python data visualisation library *matplotlib* [13], and it is fully scriptable, hence it can be used for representing the emotion annotation of single texts (e.g. for a single tweet), as well as of entire corpora (e.g. a collection of tweets), offering a tool for a proper representation of such annotated texts, which at the best of our knowledge was missing. The two-dimensional Plutchik's wheel is immediately recognisable, but it is a mere qualitative illustration. PyPlutchik introduces a quantitative dimension to this representation, making it a tool suitable for representing how much an emotion is detected in a corpus. The module accepts as an input a score $s_i \in [0, 1]$ for each of the 24 $i$ emotions in the model (8 basics emotions, 3 degrees of intensity each). Please note that, since the same text cannot express two different degrees of the same emotion, the sum of all the scores of the emotions belonging to the same branch must be less than or equal to 1. Each emotion petal is then sized according to this score. In Fig 1 we can see an example of the versatility of the PyPlutchik representation of the annotated emotions: in (i) we see a pseudo-text in which only *Joy*, *Trust* and *Sadness* have been detected; in (ii) for each emotion, the percentage of pseudo-texts in a pseudo-corpus that show that emotion; finally (iii) contains a detail of (ii), where the three degrees of intensity have been annotated separately.

Most importantly, PyPlutchik is respectful of the original spatial and aesthetic features of the wheel of emotions intended by its author. The colour code has been hard-coded in the

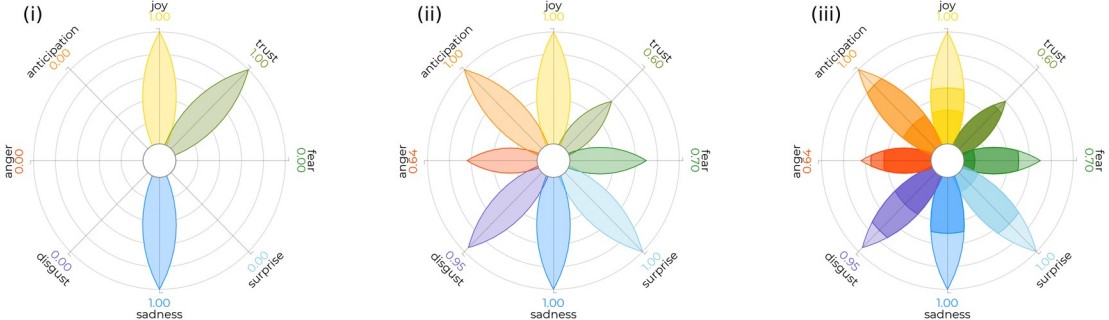

**Fig 1.** A three fold showcase of our visualisation tool on synthetic data: (i) a text where only *Joy*, *Trust* and *Sadness* have been detected; (ii) a corpus of many texts. Each petal is sized after the amount of items in the corpus that show that emotion in; (iii) same corpus as (ii), but higher and lower degrees of intensity of each emotion are expressed.

module, as it is a distinctive feature of the wheel that belongs to collective imagination. Spatial distribution of the emotion is also a standard, non customisable feature, as the displacement of each petal of the flower is non arbitrary, because it reflects a semantic proximity of close emotions, and a semantic contrariety of opposite emotions (see Sec. 2).

Representing emotions detected in texts can be hard without a proper tool, but it is a need for the many scientists that work on text and emotions. A great variety of newly available digital text has been explored in order to uncover emotional patterns; the necessity of a handy instrument to easily display such information is still unsatisfied.

In the following sections, after introducing the reader to the topic of emotion models and their applications in corpora annotation, we will focus on the Plutchik's emotion model and the current state of the art of its representations. A detailed technical explanation of the PyPlutchik module will follow, with several use cases on a wide range of datasets to help substantiating our claim that PyPlutchik outperforms other classic visualisations.

# 2 Related work

## 2.1 Visualising textual data

Visualising quantitative information associated to textual data might not be an easy task, due to "*the categorical nature of text and its high dimensionality, that makes it very challenging to display graphically*" [14]. Several scientific areas leverage visualisations techniques to extract meaning from texts, such as digital humanities [15] or social media analysis [16].

Textual data visualisations often usually provide tools for literacy and citation analysis; e.g., PhraseNet [17], Word Tree [18], Web Seer http://hint.fm/seer/, and Themail [19] introduced many different ways to generate visual overviews of unstructured texts. Many of these projects were connected to *ManyEyes*, that was launched in 2007 by Viégas, Wattenberg, and al. [20] at IBM, and closed in 2015 to be included in IBM Analytics. ManyEyes was designed as a web based community where users (mainly data analysts and visualisation designers) could upload their data to establish conversations with other users. The ambitious goal was to create a social style of data analysis so that visualisations can be tools to create collaboration and carry on discussions.

Nevertheless, all of these classic visualisation tools did not allow the exploration of more advanced textual semantic features that can be analysed nowadays due to the numerous developments of Natural Language Processing techniques; nowadays, it exists a good number of software suites—such as *Tableau*, *Microsoft PowerBI* or *Datawrapper*, just to mention a few— that give the users a chance to create very interesting, eye-catching and often complex visualizations. They all adopt a graphical user interface, with all the pros and cons that usually come with it: an intuitive and fast way to realise the majority of the most common layouts, but likely less flexible when it comes to create a more personalised visualisation. On the other hand, programming libraries and modules—such as Python's *Matplotlib* [13], *Plotly* [21] and *Bokeh* [22], or *D3.js* [23] in Javascript allow the users to create freely their own visualisations, though with a much steeper learning curve for those who are not familiar with these technologies.

The recent advancements in text technologies have enabled researchers and professional analysts with new tools to find more complex patterns in textual data. Algorithms for topic detection, sentiment analysis, stance detection and emotion detection allow us to convert very large amounts of textual data to actionable knowledge; still, the outputs of such algorithms can be too hard to consume if not with an appropriate data visualisation [24]. During the last decade, many works have been carried out to fill this gap in the areas of (hierarchical) topic visualisation [25–28], sentiment visualisation (a comprehensive survey can be found in [29]), online hate speech detection [30], stance detection [31] and many more.

Our work lies within the domain of visualisation of emotions in texts, as we propose a novel Python implementation of Plutchik's wheel of emotions.

## 2.2 Understanding emotions from texts

In the last few years, many digital text sources such as social media, digital libraries or television transcripts have been exploited for emotion-based analyses. To mention just a few examples, researchers have studied the display of emotions in online social networks like Twitter [32–35] and Facebook [36–38], in literature corpora [39, 40], in television conversations [41], in dialogues excerpts from call centres conversations [42], in human-human video conversations [43]. In [44] the authors, after developing a methodology (the so-called TFMN, *textual forma mentis networks*) that exploits complex networks to extract the semantic framework of concepts from texts, resort to the Plutchik's model to analyse it.

Among categorical emotion models, Plutchik's wheel of emotions is one of the most popular. Categorical (or discrete) emotions model root to the works of Paul Ekman [1], who first recognised six basic emotions universal to the human kind (*Anger*, *Disgust*, *Fear*, *Happiness*, *Sadness*, *Surprise*). Although basicality of emotions is debated [45], categorical emotions are very popular in natural language processing research, because of their practicality in annotation. In recent years many other categorical emotion models have been proposed, each with a distinctive set of basic emotions: the model first proposed by James [2] presents 6 basic emotions, Plutchik's model 8, Izard's model [3] 12, Lazarus et al. [4] model 15, Ekman's extended model [46] 18, Cowen et al. [47] 27. Parrott [48] proposed a tree-structured model with 6 basic emotions on a first level, 25 on a second level and more than one hundred on a third level. Susanto et al. in [49] propose a revisited version of the *hourglass of emotions* by Cambria et al. [50], an interesting model that moves from Plutchik's one by positioning emotions in an hourglass-shaped design.

However, annotation of big corpora of texts is easier if labels are in a small number, clearly distinct from each other; on the other hand, a categorical classification of complex human emotions into a handful of basic labels may be limiting.

Plutchik's model's popularity is probably due to a peculiar characteristic. In its wheel of emotions, there are 8 basic emotions (*Joy*, *Trust*, *Fear*, *Surprise*, *Sadness*, *Disgust*, *Anger* and *Anticipation*) with three intensity degrees each, as shown in Table 1 and in Fig 2. Even if each emotion is a category on its own, emotions are related each other by their spatial displacement. In fact, four emotions (*Anger*, *Anticipation*, *Joy*, *Trust*) are respectively opposed to the other four (*Fear*, *Surprise*, *Sadness*, *Disgust*); for instance, *Joy* is the opposite of *Sadness*, hence it is displayed symmetrically with respect to the centre of the wheel. When elicited together, two emotions raise a *dyad*, a complex emotion. Dyads are divided into primary (when triggered by two adjacent emotions), secondary (when triggered by two emotions that are 2 petals away), tertiary (when triggered by two emotions that are 3 petals away) and opposite (when triggered by opposite emotions). A comprehensive diagram of emotion combination and elicited dyads is represented in Fig 3. This mechanism allows to annotate a basic set of only 8 emotions, while triggering eventually up to 28 more complex nuances, that better map the complexity of human emotions. When representing corpora annotated following Plutchik's model, it is important then to highlight spatial adjacency or spatial opposition of emotions in a graphical way. We will refer to these feature as *semantic proximity* and *semantic opposition* of two emotions.

From a data visualisation point of view, PyPlutchik's closest relatives can be found in bar plots, radar plots and Windrose diagrams. Bar plots correctly display the quantitative representation of categorical data, while radar plots (also known as spider plot) correctly displace

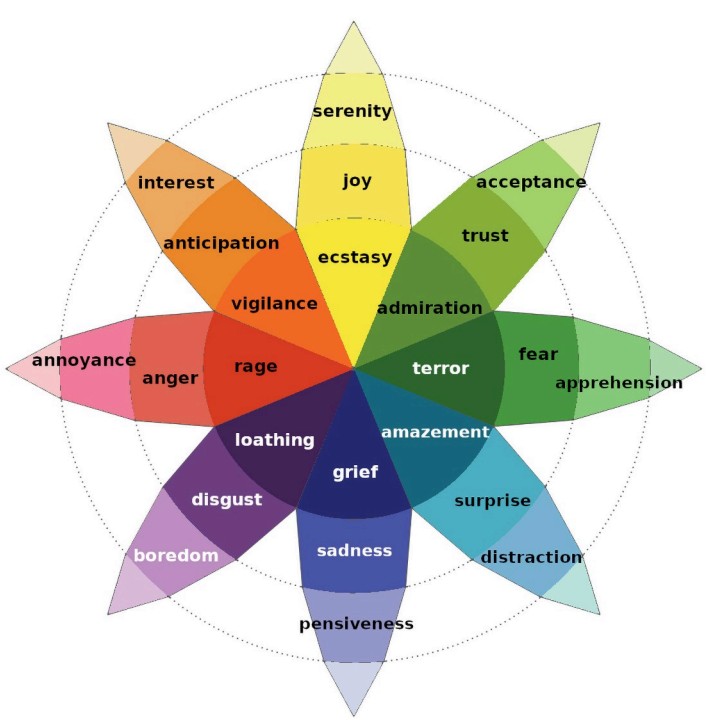

**Fig 2. Plutchik's wheel of emotions.** Each petal is partitioned in three degrees of intensity, from the most intense (the most internal section) to the least intense (the most external section).

elements in a polar coordinate system, close to the original Plutchik's one. Windrose diagrams combine both advantages, displaying categorical data on a polar coordinate system. PyPlutchik is inspired to this representation, and it adapts this idea to the collective imagination of Plutchik's wheel of emotion graphical picture.

## 2.3 Representing Plutchik's emotions wheel

If we skim through the related literature, we notice that many papers needed to display the distribution of emotions in a corpus, and without a dedicated tool they all settled for a practical but sub-optimal solution. In some way, each of the following representations does not respect the standard spatial or aesthetic features of Plutchik's wheel of emotions:

- **Tables**, as used in [51–56]. Tables are a practical way to communicate exact amounts in an unambiguous way. However, tables are not a proper graphical display, so they miss all the features of the original wheel of emotions: there is not a proper colour code and both semantic proximity and semantic opposition are dismantled. Confronted with a plot, texts are harder to read: plots deliver the same information earlier and easier.

- **Bar plots**, as used in [57–63]. Bar plots are a traditional option to allow numerical comparisons across categories. In this domain, each bar would represent how many times a emotion is shown in a given corpus. However, bar plots are sub-optimal for two reasons. Firstly, the spatial displacement of the bars does not reflect semantic opposition of two emotions, that are opposites in the Plutchik's wheel. Secondly, Plutchik's wheel is circular, meaning that there is a semantic proximity between the first and the last of the 8 emotions branches,

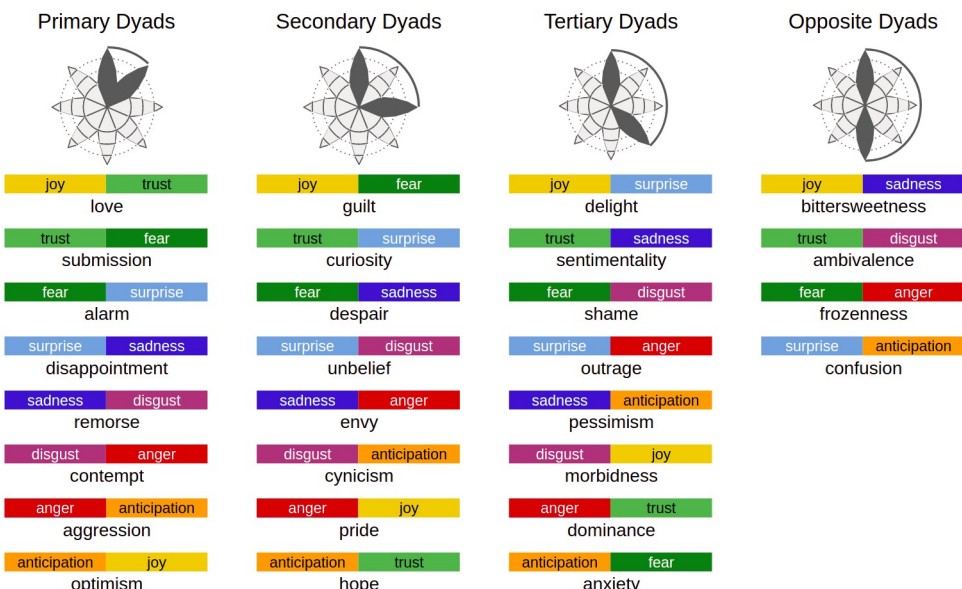

**Fig 3. Diagram of Plutchik's dyads.** When two emotions are elicited together, they trigger the corresponding primary dyad if they are just one petal apart, a secondary dyad if they are two petal distant each other, a tertiary dyad if they are three petal distant, an opposite dyad if they are on the opposite side of the flower.

which is not represented in a bar plot. PyPlutchik preserves both semantic opposition and semantic proximity: the mass distribution of the ink in Fig 14(i and vi), for instance, immediately communicates of a positive corpus, as positive emotions are way more expressed than their opposites.

- **Pie charts**, as used in [55, 64–67]. Pie charts are a better approximation of the Plutchik's wheel, as they respect the colour code and they almost respect the spatial displacement of emotions. However, the actual displacement may depend on the emotion distribution: with a skewed distribution toward one or two emotions, all the remaining sectors may be shrunk and translated to a different position. Pie charts do not guarantee a correct spatial positioning of each category. There is also an underlying conceptual flaw in pie charts: they do not handle well items annotated with more than one tag, in this case texts annotated with more than one emotion. In a pie chart, the sum of the sectors' sizes must equal the number of all the items; each sector would count how many items fall into a category. If multiple annotation on the same item are allowed, the overall sum of sectors' sizes will exceed the number of actual items in the corpus. Null-annotated items, i.e. those without a noticeable emotion within, must be represented as a ninth, neutral sector. PyPlutchik handles multi-annotated and null-annotated items: for instance, Fig 1(ii) shows a pseudo-corpus where *Anger* and *Disgust* both are valued one, because they appear in 100% of the pseudo-texts within. Fig 1(i) shows a text with several emotions missing.

- **Heatmaps**, as used in [68, 69]. Both papers coded the intensity of the 8 basic emotions depending on a second variable, respectively time and principal components of a vectorial representation of texts. Although heatmaps naturally fit the idea of an intensity score at the crossroad of two variables, the final display are sub-optimal in both cases, because they fail to preserve both the Plutchik's wheel's colour code and spatial displacement of emotions. As described in Sect. 3, PyPlutchik can be easily scripted for reproducing small-multiples. In

Sect. 5 we provide an example of a small-multiple, displaying the evolution of the distribution of emotions in a corpus over time.

- **Scatter plots**, as used in [70]. Scatter plot are intended to display data points in a two- or three-dimensional space, where each axis maps a continuous variable. In [70], x-axis represents the rank of each emotion on each of the three corpora they analyse, thus producing a descending sorting of emotion labels. This choice was probably made in order to have three descending, more readable series of scatters on the plot. However, this representation breaks both the colour code and the spatial displacement of emotions. PyPlutchik can be easily scripted for a side-by-side comparison of more than one corpus (see Sect. 3), allowing readers to immediately grasp high level discrepancies.

- **Line plots**, as used in [71]. As well as scatter plots, line plots are appropriate for displaying a trend in a two-dimensional space, where each dimension maps a continuous variable. It is not the case of discrete emotions. Authors plotted the distribution of each emotion over time as a separate line. They managed to colour each line with the corresponding colour in the Plutchik's wheel, reporting the colour code in a separate legend. As stated before in similar cases, this representation breaks the semantic proximity (opposition) of close (opposite) emotions. Again, in Sect. 3 we provide details about how to script PyPlutchik to produce a small-multiple plot, while in Sec. 5 we showcase the distribution of emotions by time on a real corpus.

- **Radar plots**, as used in [72–74]. Radar plots, a.k.a. Circular Column Graphs or Star Graphs, successfully preserve spatial proximity of emotions. Especially when the radar area is filled with a non-transparent colour, radars correctly distribute more mass where emotions are more expressed, giving to the reader an immediate sense of how shifted a corpus is against a neutral one. However, on a minor note, continuity of lines and shapes do not properly separate each emotion as a discrete objects per se. Furthermore, radars do not naturally reproduce the right colour code. Lastly, radars are not practical to reproduce stacked values, like the three degrees of intensity in Fig 1 (i). Of course, all of these minor issues can be solved with an extension of the basic layout, or also adopting a **Nightingale Rose Chart** (also referred as Polar Area Chart or Windrose diagram), as in [35, 75]. However, the main drawback with radar plots and derivatives is that semantic opposition is lost, and we do not have a direct way to represents dyads and their occurrences. PyPlutchik, conversely, has been tailored on the original emotion's wheel, and it naturally represents both semantic proximity and opposition, as well as the occurrences of dyads in our corpora (see Sect. 4).

## 3 Visualising primary emotions with PyPlutchik

PyPlutchik is designed to be integrated in data visualisation with the Python library matplotlib. It spans the printable area in a range of $[-1.6, 1.6]$ inches on both axes, taking the space to represent a petal of maximum length 1, plus the outer labels and the inner white circle. Each petal overlaps on one of the 8 axis of the polar coordinate system. Four transversal minor grid lines cross each axis, spaced of 0.2 inches each, making it a visual reference for a quick evaluation of the petal size and for a comparison between non adjacent petals. Outside the range $[0, 1]$, corresponding to each petal, two labels represent the emotion and the associated numerical score. Colour code is strictly hard-coded, following Plutchik's wheel of emotions classic

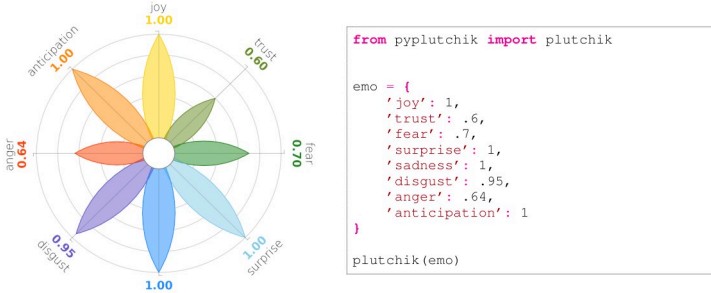

**Fig 4. Plutchik's wheel generated by code on the right.** Each entry in the Python *dict* is a numeric value ∈ [0, 1].

representation. PyPlutchik can be used either to plot only the 8 basic emotions, or to show the full intensity spectrum of each emotion, assigning three scores for the three intensity levels. In the latter case, each petal is divided into three sections, with colour intensity decreasing from the centre. In both cases PyPlutchik accepts as input a *dict* data structure, with exactly 8 items. Keys must be the 8 basic emotions names. *dict* is a natural Python data structure for representing JSON files, making PyPlutchik an easy choice to display JSONs. In case of basic emotions only, values in the *dict* must be numeric ∈ [0, 1], while in case of intensity degrees they must be presented as an *iterable* of length three, whose entries must sum to maximum 1. Figs 4 and 5 show how straightforward it is to plug a *dict* into the module to obtain the visualisation. Furthermore, PyPlutchik can be used to display the occurrences of primary, secondary, and tertiary dyads in our corpora. This more advanced feature will be described in Sect. 4.

Due to the easy integration with Python basic data structures and the matplotlib library, PyPlutchik is also completely scriptable to display several plots side by side as small-multiple. Default font family is sans-serif, and text is printed with light weight and size 15 by default. However, it is possible to modify these features by the means of the corresponding parameters (see the documentation the Pyplutchik documentation is available in the Github repository at https://github.com/alfonsosemeraro/pyplutchik/blob/master/Documentation.md for a comprehensive and always up-to-date explanation of parameters and features of the module). These features can be also changed with standard matplotlib syntax. The polar coordinates beneath petals and the labels outside can be hidden: this feature leaves only the flower on screen, improving visibility of small flowers in small-multiple plots. Also the petals aspect can be modified, by making them thinner or thicker. Fig 6 shows a small-multiple, with hidden

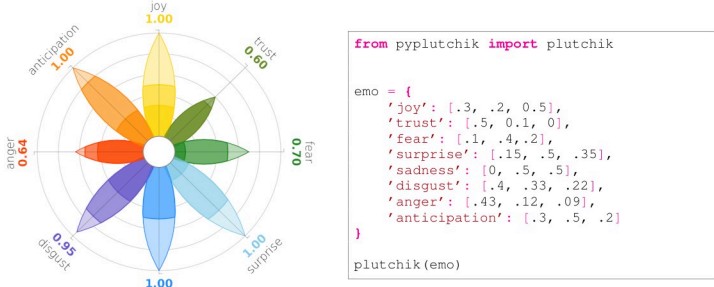

**Fig 5. Plutchik's wheel generated by code on the right.** Each entry in the Python *dict* is a three-sized array, whose sum must be ∈ [0, 1].

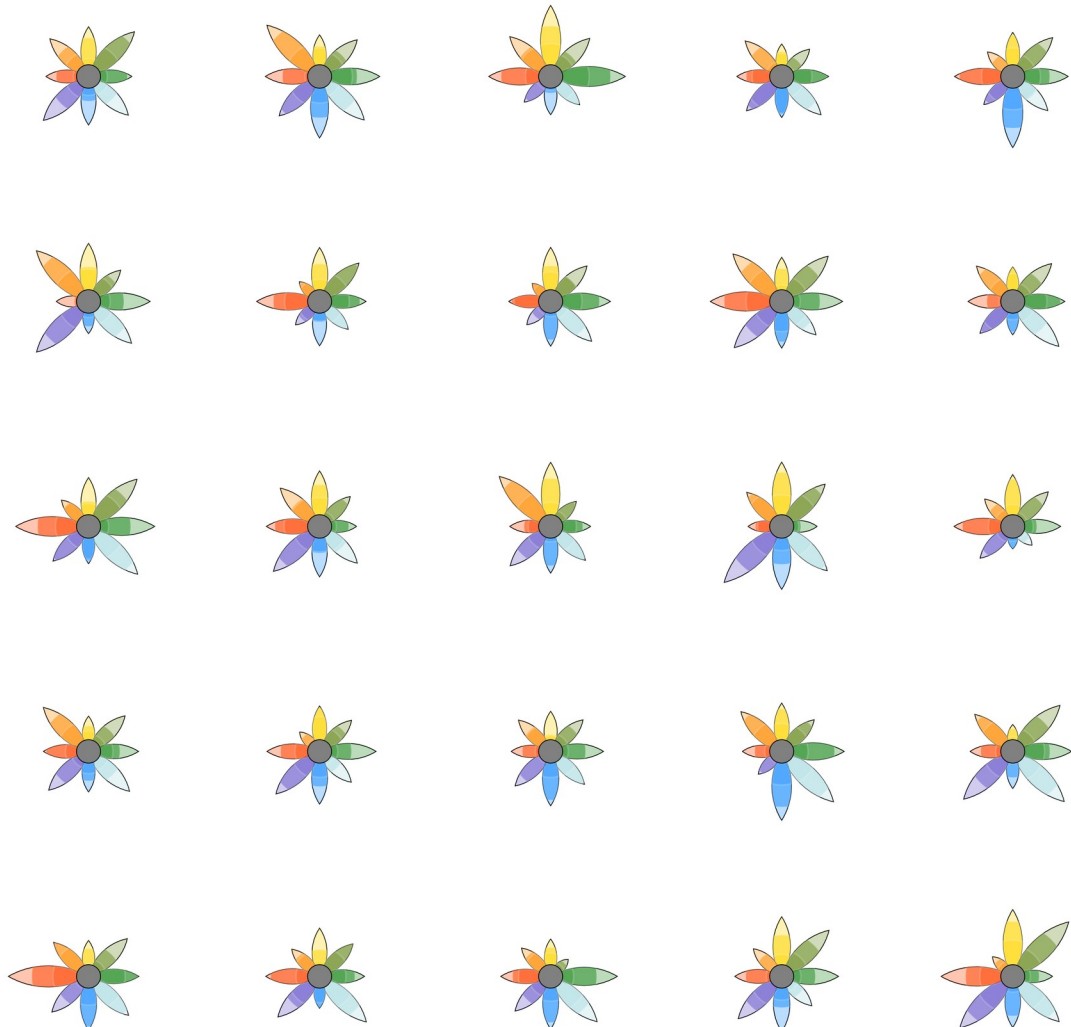

**Fig 6. Small-multiple of a series of Plutchik's wheel built from synthetic data.** Polar coordinates beneath the flowers and labels around have been hidden to improve the immediate readability of the flowers, resulting in a collection of emotional fingerprints of different corpora.

polar coordinates and labels, computed on synthetic random data that have been artificially created only for illustrative purposes. Code for this kind of representation is in the code documentation, as well as a short overview on the parameters that make possible to reproduce the aforementioned variations.

As a further customisation option, we allow the user to select a set of petals to be highlighted. This selective presentation feature follows a focus-plus-context [76] approach to the need of emphasising those emotions that might be more distinctive, according to the case under consideration. We chose to apply a focus-plus-context visualisation by filling petals' areas selectively, without adopting other common techniques, as with fish-eye views [77], in order to avoid distortions and to preserve the spatial relations between the petals. It is possible to select a list of main emotions, and to show a colored petal and all three intensity scores for each emotion in the list, while for the others it will display a grey petal and the cumulative scores only, also in grey. We showcase this feature in Fig 7 and in the code documentation.

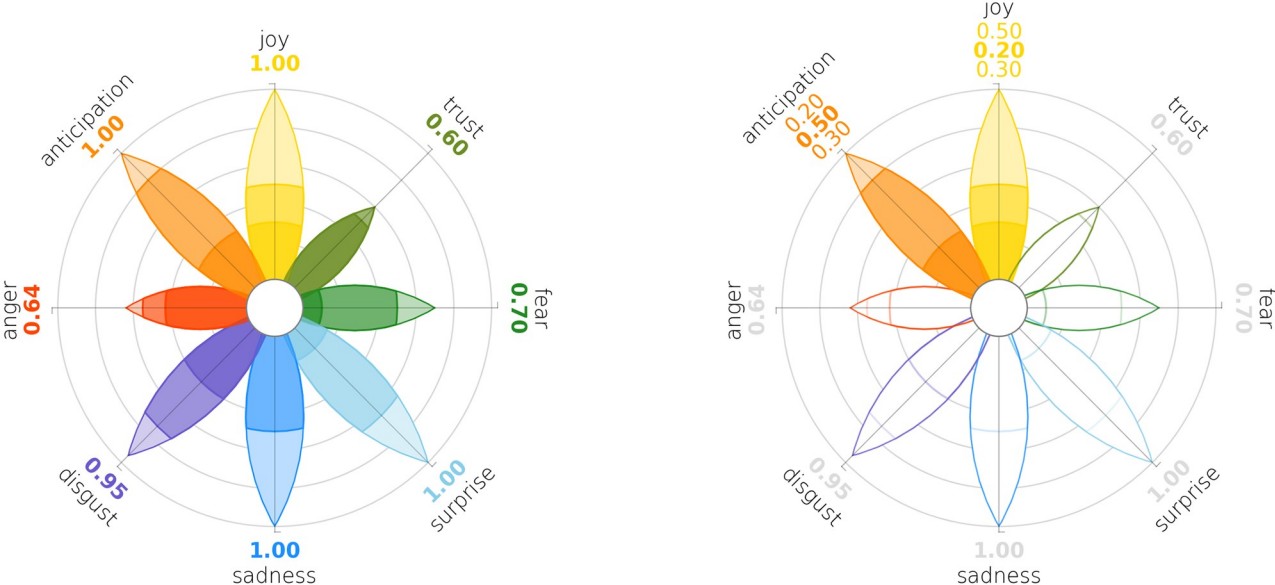

**Fig 7. A side-by-side comparison between the synthetic plot of Fig 1(iii) and an almost identical wheel, but with only two emotions highlighted.** We highlighted and displayed the three intensity scores of *Anticipation* and *Joy*.

## 4 Showing dyads with PyPlutchik

Dyads are a crucial feature in Plutchik's model. As explained in Sec. 1, the high flexibility of the model derives also from the spatial disposition of the emotions. Primary emotions can combine with their direct neighbours, forming primary dyads, or with emotions that are two or three petals away, forming respectively secondary and tertiary dyads. Opposite dyads can be formed as well, by combining emotions belonging to opposite petals. This feature dramatically enriches the spectrum of emotions, beyond the primary ones; enriching the spectrum increases the complexity of the models, also contributing to mitigate, at least partially, the phenomenon of cultural variation discussed in Sec. 1. Therefore, a comprehensive visualisation of Plutchik's model must offer a way to visualise dyads.

The design of such a feature is non trivial. Indeed, while the flower of primary emotions is inherent to the model itself, no standard design is provided to visualise dyads. For our implementation we decided to stick with the flower-shaped graphics, in order not to deviate too much from the original visualisation philosophy. Examples that show all levels of dyads can be seen in Figs 14 and 15. While the core of the visual remains the same, a few modifications are introduced. In more detail:

- the radial axes are progressively rotated by 45 degrees in each level, to enhance the spatial shift from primary emotions to dyads;

- the petals are two-tone, according to the colours of the primary emotions that define each dyad;

- a textual annotation in the center gives an indication of what kind of dyad is represented: "1" for primary dyads, "2" for secondary dyads, "3" for tertiary dyads, "opp." for opposite dyads.

- while the dyads labels all come in the same colour (default is black), an additional circular layer has been added in order to visualise the labels and the colours of the primary emotions that define each dyad.

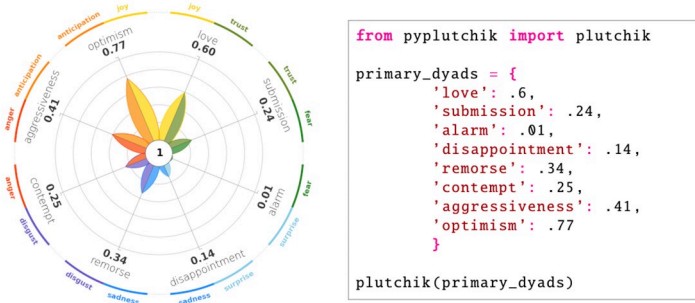

**Fig 8. Primary dyads' wheel generated by code on the right.** Each entry in the Python *dict* is a numeric value ∈ [0, 1].

This last feature is particularly useful to give the user an immediate sense of the primary emotions involved in the formation of the dyad. Fig 8 provides an example of the wheel produced if the user inputs a *dict* containing primary dyads instead of emotions. PyPlutchik automatically checks for the kind of input wheel and for its coherence: specifically, the module retrieves an error if the input dictionary contains a mix of emotions from different kind of dyads, as they cannot be displayed on the same plot. In Fig 9 we show a representation of basic emotions, primary dyads, secondary dyads, tertiary dyads and opposite dyads, based on synthetic data. This representation easily conveys the full spectrum of emotions and their combinations according to Plutchik's model, allowing for a quick but in-depth analysis of emotions detected in a corpus.

## 5 Case studies

We now showcase some useful examples of data visualisation using PyPlutchik. We argue that PyPlutchik is more suitable than any other graphical tool to narrate the stories of these examples, because it is the natural conversion of the original qualitative model to a quantitative akin, tailored to visually represent occurrences of emotions and dyads in an annotated corpus.

### 5.1 Amazon office products reviews

As a further use case we exploit a dataset of products review on Amazon [78]. This dataset contains almost 142.8 millions reviews spanning May 1996—July 2014. Products are rated by the customers on a 1-5 stars scale, along with a textual review. Emotions in these textual reviews have been annotated using the Python library NRCLex [79], which checks the text against a lexicon for word-emotion associations; we do not have any ambition of scientific accuracy of the results, as this example is meant for showcasing our visualisation layouts.

In Fig 10 we plot the average emotion scores in a sample of reviews of office products, grouped by the star-ratings. We can sense a trend: moving from left to right, i.e. from low-rates to high-rates products, we see the petals in the top half of the flower slowly growing in size at the expense of the bottom half petals. The decreasing effect is particularly visible in *Fear*, *Anger* and *Disgust*.

This visualisation is effective in communicating the increasing satisfaction of the customers; nevertheless, this improvement is very gradual and can hardly be noticed by comparing to subsequent steps. As we can see from Fig 11(a), it is much more evident if we compare one-star-rated products to five-star-rated product reviews. The selective presentation feature of our module (Fig 11(b)) is a good way to enhance this result: it allows to put emphasis on the

**Fig 9. Representation of emotions and primary, secondary, tertiary and opposite dyads.** The data displayed is random.



**Fig 10. Average emotion scores in a sample of textual reviews of office products on Amazon.** Rating of products goes from one star (worst) to five (best). On the left, emotions detected in negative reviews (one star), on the right the emotions detected in positive reviews (five star). While positive emotions stay roughly the same, negative emotions such *Anger*, *Disgust* and *Fear* substantially drop as the ratings get higher.

desired emotions without losing sight of the others, that are left untouched in their size or shape but are overshadowed, deprived of their coloured fill.

### 5.2 Emotions in IMDB movie synopses

In Fig 12 is shown the emotion detected in the short synopses from the top 1000 movies on the popular website IMDB (Internet Movie Data Base). Data is an excerpt of only four genres (namely Romance, Biography, Mystery and Animation) taken from Kaggle [80], and emotions have been annotated again with the Python library NRCLex. As in the previous case, both the dataset and the methodology are flawed for the task: for instance the synopsis of the movie may describe a summary of the main events or of the characters, but with detachment; the library lexicon may not be suited for the movie language domain. However, data here is presented for visualisation purposes only, and not intended as a contribution in the NLP area.

Romance shows a slight prominence of positive emotions over negative ones, especially over *Disgust*. The figure aside represents the Biography genre, and it is immediately distinctive for the high *Trust* score, other than higher *Fear*, *Sadness* and *Anger* scores. While high *Trust* represents the high admiration for the subject of the biopic, the other scores are in line with Propp's narration scheme [81], where the initial equilibrium is threatened by a menace the hero is called to solve. A fortiori, Mystery's genre conveys even more *Anger* and more *Sadness* than Biography, coupled with a higher sense of *Anticipation* and a very high score for *Fear*, as expected. Last, the Animation genre arouses many emotions, both positive and negative, with high levels of *Joy*, *Fear*, *Anticipation* and *Surprise*, as a children cartoon is probably supposed to do. Printed together, these four shapes are immediately distinct from each other, and they return an intuitive graphical representation of each genre's peculiarities. Shapes are easily recognisable as positive or negative, bigger petals are predominant and petals' sizes are easy to compare with the aid of the thin grid behind them.

Data represented in Fig 12 is a larger excerpt of the same IMDB dataset, which covers 21 genres. The whole dataset gives us the chance to show a small-multiple representation without visible coordinates, as described in Sect. 3: we plotted in Fig 13 the most common 20 genres of movies within the top 1000, 5 by row. We hid the grid and the labels, leaving the flower to speak for itself. Data represented this way is not intended to be read with exactness on numbers. Instead, it is intended to be read as an overall overview on the corpus. Peculiarities, outliers and one-of-a-kind shapes catch the eye immediately, and they can be accurately scrutinised later with a dedicated plot that zooms into the details. For instance, the Film-Noir genre contains only a handful of movies, whose synopses are almost always annotated as emotion-heavy. The resulting shape is a clear outlier in this corpus, with extremely high scores on 5 of 8

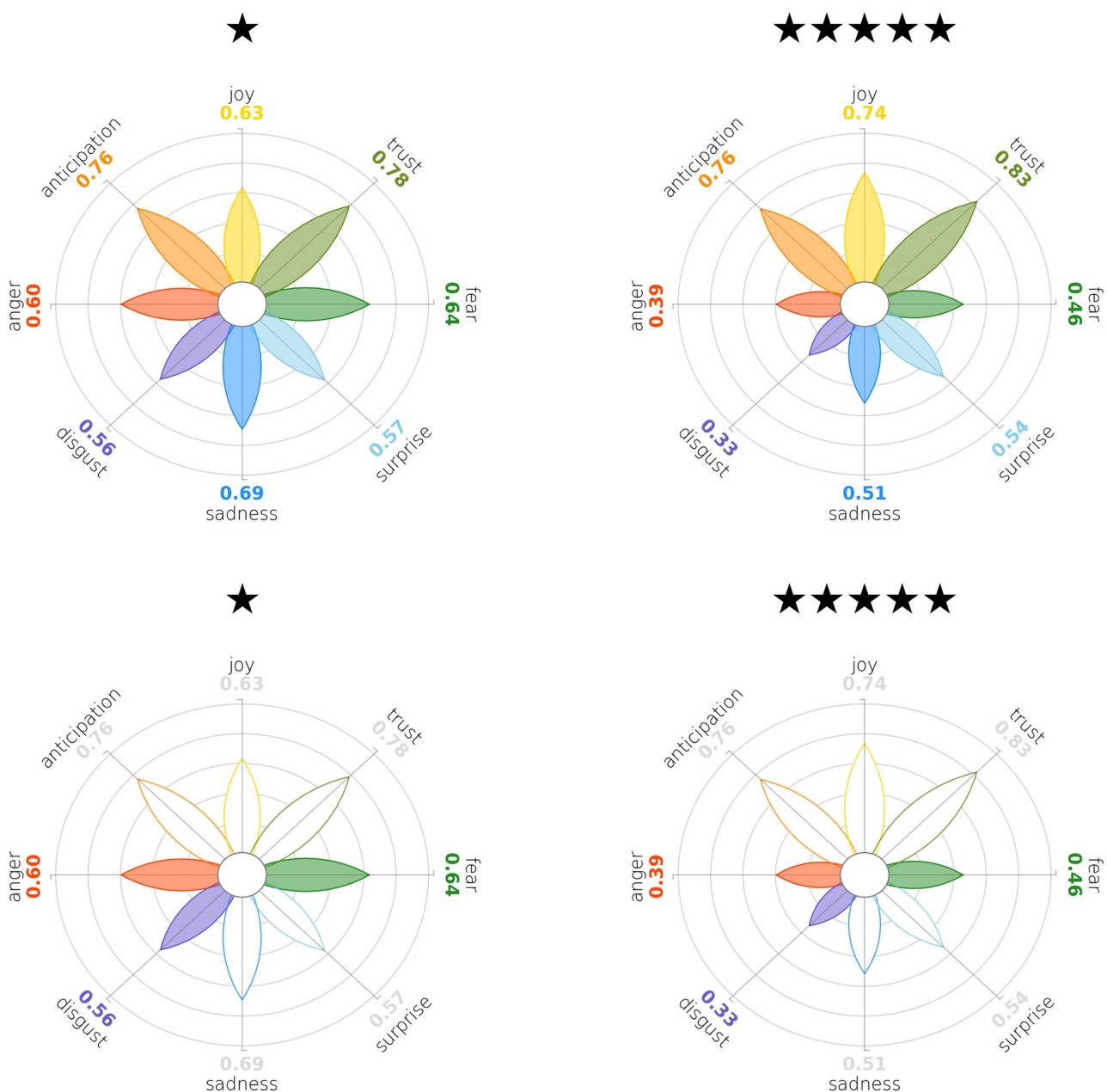

**Fig 11. Focus-plus-context: The selective presentation feature of PyPlutchik allows to put emphasis on some particular emotions, without losing sight of the others; we can compare different subgroups of the same Amazon corpus placing our visualisations side-by-side, and highlighting only** *Anger*, *Disgust* **and** *Fear* **petals, to easily spot how these negative emotions are under represented in 5-stars reviews than in 1-star reviews.**

emotions. Thrillers and Action movies share a similar emotion distribution, while Music and Musical classify for the happiest.

### 5.3 Trump or Clinton?

In Figs 14 and 15 we visualise the basic emotions and dyads found in tweets in favour and against Donald Trump and Hillary Clinton, the 2016 United States Presidential Elections

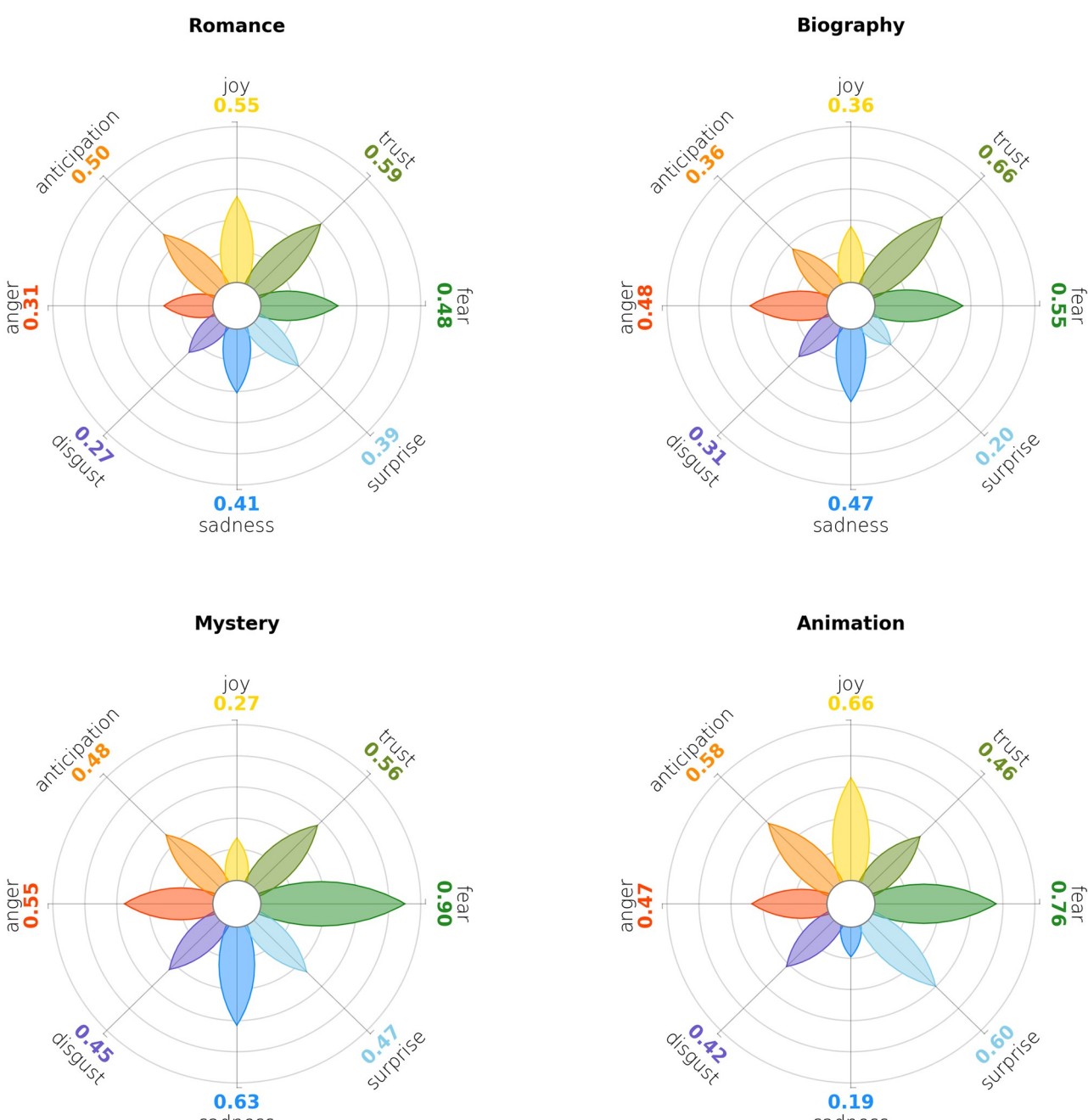

**Fig 12. Emotions in the synopses of the top 1000 movies in the IMDB database, divided by four genres.** The shapes are immediately distinct from each other, and they return an intuitive graphical representation of each genre's peculiarities.

principal candidates. Data is the training set released for a SemEval 2016 task, namely a corpus of annotated stances, sentiments and emotions in tweets [82]. Each candidate is represented in both plots on a different row, and each row displays five subplots, respectively basic emotions, primary dyads, secondary dyads, tertiary dyads and opposite dyads. Tweets supporting either Trump or Clinton present higher amounts of positive emotions (Fig 14(i) and 14(vi)), namely

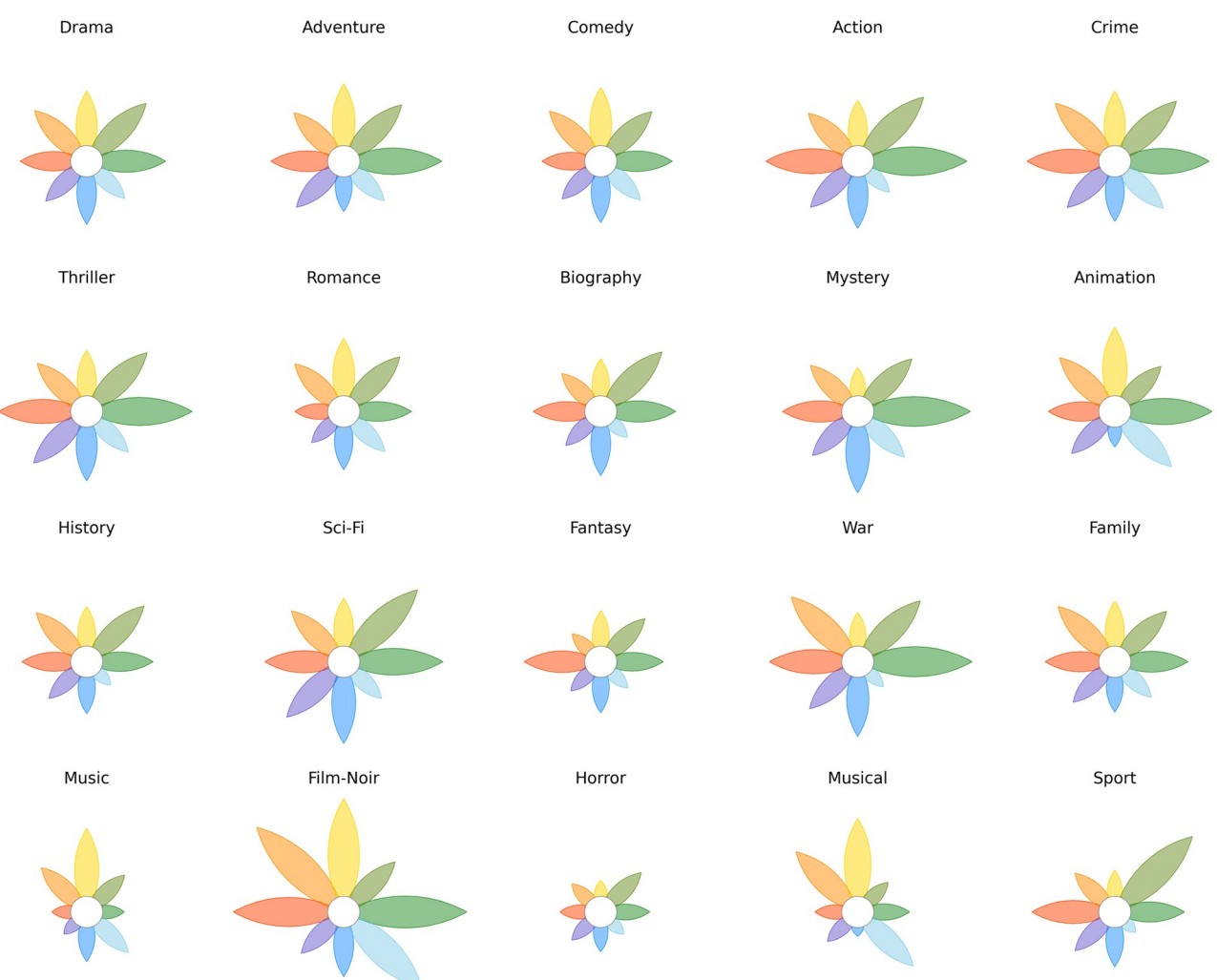

**Fig 13. Emotions in the synopses of the 20 most common movie genres in the IMDB database.** Coordinates, grids and labels are not visible: this is an overall view of the corpus, meant to showcase general trends and to spot outliers that can be analysed at a later stage, in dedicated plot.

*Anticipation*, *Joy* and *Trust*, and from lower to no amounts of negative emotions, especially *Sadness* and *Disgust*. On the contrary, tweets critical of each candidate (Fig 15(i) and 15(vi)) show high values of *Anger*, coupled with *Disgust*, probably in the form of disapproval.

There are also significant differences between the two candidates. Donald Trump collects higher levels of *Trust* and *Anticipation* from his supporters than Hillary Clinton, possibly meaning higher expectations from his electoral base. Users that are skeptical of Hillary Clinton show more *Disgust* towards her than Donald Trump's opponents towards him.

Besides basic emotions, PyPlutchik can display the distribution of dyads as well, as described in Sec. 4. Dyads allow for a deeper understanding of the data. We can see how the tweets against the presidential candidates in Fig 15 are dominated by the negative basic emotion of *Anger*, with an important presence of *Disgust* and *Anticipation* (subplots (i) and (vi)); the dominant primary dyad is therefore the co-occurence of *Anger* and *Disgust* (subplot (ii)), i.e. the primary dyad *Contempt*, but not *Aggressiveness*, the primary dyad formed by *Anger* and *Anticipation*: the latter rarely co-occurs with the other two, which means that expectations and

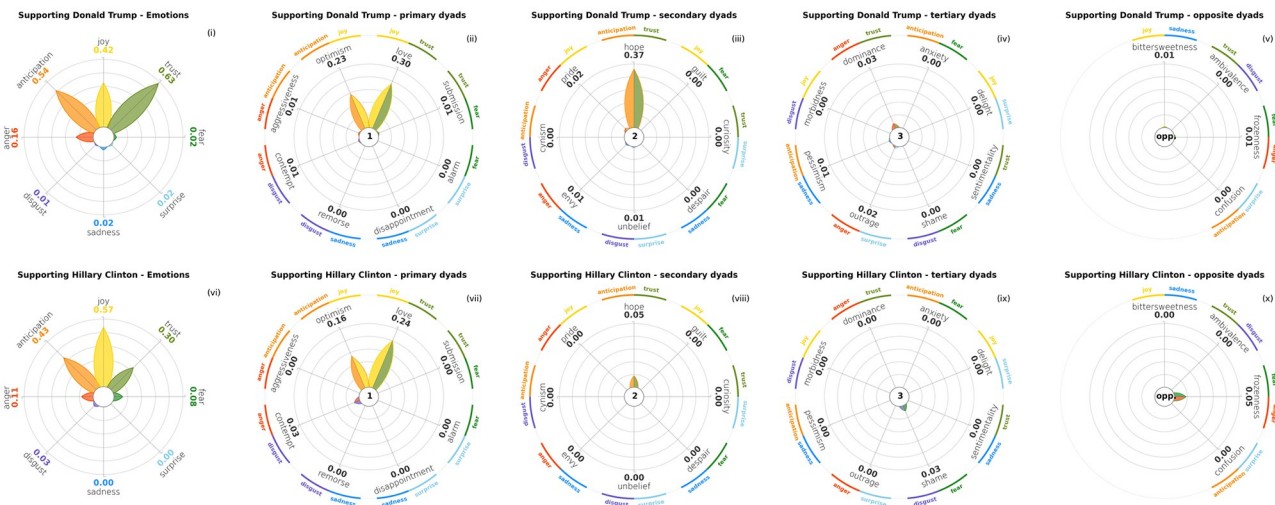

**Fig 14. Tweets in favour of Donald Trump and Hillary Clinton from the 2016 StanceDetection task in SemEval.** From left to right: basic emotions, primary dyads, secondary dyads, tertiary dyads and opposite dyads for both candidates (Donald Trump on the first row, Hillary Clinton on the second one). Despite the high amounts of *Anticipation*, *Joy* and *Trust* for both the candidates, which result in similar primary dyads, there is a significant spike on the secondary dyad *Hope* among Trump's supporters that is not present in Clinton's supporters.

contempt are two independent drives in such tweets. The other dyads are relatively scarcer as we progress on the secondary and tertiary level (subplots (iii)-(v)). The supporting tweets in Fig 14 are characterised by positive emotions, both in the primary flower and in the dyads, with these again being a reflection of the co-occurrence of the most popular primary emotions. Although *Anticipation*, *Joy* and *Trust* are present in different amounts, primary dyads *Optimism* and *Love* do occur in a comparable number of cases (subplot (ii)). Interestingly, the pro-Trump tweets show a remarkable quantity of *Hope*, the secondary dyad that combines

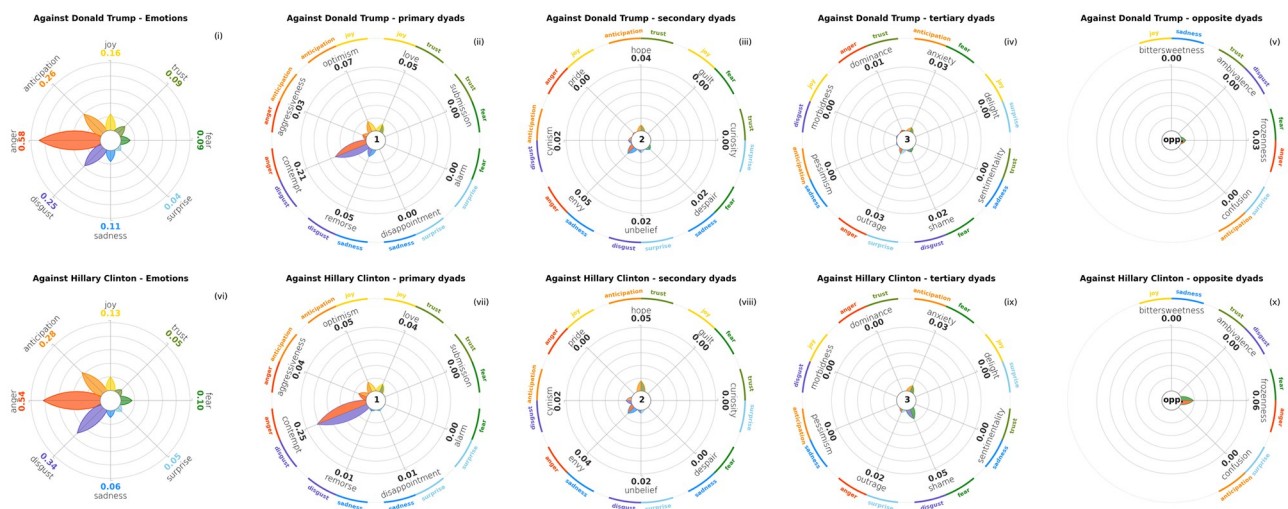

**Fig 15. Similarly to Fig 14, here are shown the emotions captured in the tweets against Donald Trump and Hillary Clinton from the 2016 StanceDetection task in SemEval.** We see a clear prevalence of negative emotions, particularly *Anger* and *Disgust*. This combination is often expressed together, as can be seen from the primary emotions plots (ii and vii), where there is a spike in *Contempt*.

*Anticipation* and *Trust*, suggesting that Trump's supporters expressed towards him all the three dominant basic emotions together more often that Clinton's supporters did.

Generally speaking, we notice that there are not many dyads expressed in the tweets. We can ascribe this to many factors: first and foremost, a dataset annotated *only* on primary emotions and not also explicitly on the dyads will naturally show less dyads, since their presence will depend only on the casual co-occurrence of primary emotions. For this reason we resized the petal lengths in Fig 14(ii)–14(v) and 14(vii)–14(x) with a different scale than (i) and (v), and the same applies for Fig 15. For instance, *Trust* petal in Fig 14(i) is almost to the maximum length, with a score equal to 0.63; *Hope* petal in (iii) is also almost at the maximum length, but with a score equal to 0.37. Both figures are therefore designed for a visual comparison of the petals within the same subplot, but not between two different subplots, due to the lack of consistency in the scale. However, this does not concern us, since our current purpose is to showcase the potential of PyPlutchik: with this regard, we note that we were immediately able to track the unexpected presence of the secondary dyad *Hope*, that stood out among the others.

## 6 Conclusion

The increasing production of studies that explore the emotion patterns of human-produced texts often requires dedicated data visualisation techniques. This is particularly true for those studies that label emotions according to a model such as Plutchik's one, which is heavily based on the principles of semantic proximity and opposition between pairs of emotions. Indeed, the so-called *Plutchik's wheel* is inherent to the definition of the model itself, as it provides the perfect visual metaphor that best explains this theoretical framework.

Nonetheless, by checking the most recent literature it appears evident that too often this aspect is neglected, instead entrusting the visual representation of the results to standard, suboptimal solutions such as bar charts, tables, pie charts. We believe that this choice does not do justice neither to the said studies, nor to Plutchik's model itself, and it is mainly due to the lack of an easy, plug-and-play tool to create adequate visuals.

With this in mind we introduced PyPlutchik, a Python module for the correct visualisation of Plutchik's emotion traces in texts and corpora. PyPlutchik fills the gap of a suitable tool specifically designed for representing Plutchik's model of emotions. Most importantly, it goes beyond the mere qualitative display of the Plutchik's wheel, that lacks a quantitative dimension, allowing the user to display also the quantities related to the emotions detected in the text or in the corpora. Moreover, PyPlutchik goes the extra mile by implementing a new way of visualising primary, secondary, tertiary and opposite dyads, whose presence is a distinctive and essential feature of the Plutchik's model of emotions.

This module is built on top of the popular Python library matplotlib, its APIs being written in a matplotlib style. PyPlutchik is designed for an easy plug and play of JSON files, and it is entirely scriptable. It is designed for single plots, pair-wise and group-wise side-by-side comparisons, and small-multiples representations. The original Plutchik's wheel of emotions displace the 8 basic emotions according to proximity and opposition principles; the same principles are respected in PyPlutchik layout.

It should be noted that the PyPlutchik module is meant to work only with data that was classified—manually or automatically—according to the Plutchik's model of emotions. In general, it is never advisable to represent data on a different scheme from the one used for its classification. Most of the models are radically different from each other, both in the underlying assumptions as well as in the categories scheme; even when they are somewhat similar, there might be key differences that should be kept into account. As for the Plutchik model in particular, to the best of our knowledge it does not exist a one-to-one correspondence with any

other discrete emotional model, as well as no methods to go from a continuous emotional space to a discrete one. For these reasons, PyPlutchik should be used only with data annotated following the Plutchik's scheme of annotation.

As we pointed out, currently there are thousands of empirical works on Plutchik's model of emotions. Many of these works need a correct representation of the emotions detected or annotated in data. It is our hope that our module will help the scientific community by providing them with an alternative to the sub-optimal representations of Plutchik's emotion currently available in literature.

## Author Contributions

**Conceptualization:** Alfonso Semeraro, Salvatore Vilella, Giancarlo Ruffo.

**Data curation:** Alfonso Semeraro, Salvatore Vilella.

**Formal analysis:** Alfonso Semeraro.

**Methodology:** Alfonso Semeraro, Salvatore Vilella, Giancarlo Ruffo.

**Project administration:** Giancarlo Ruffo.

**Software:** Alfonso Semeraro.

**Supervision:** Giancarlo Ruffo.

**Validation:** Alfonso Semeraro, Salvatore Vilella.

**Visualization:** Alfonso Semeraro, Salvatore Vilella, Giancarlo Ruffo.

**Writing – original draft:** Alfonso Semeraro.

**Writing – review & editing:** Salvatore Vilella, Giancarlo Ruffo.

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
