## [Decision Letter · Decision Letter 0]

8 Jul 2021

PONE-D-21-18896

PyPlutchik: visualising and comparing emotion-annotated corpora

PLOS ONE

Dear Dr. Vilella,

Thank you for submitting your manuscript to PLOS ONE. After careful consideration, we feel that it has merit but does not fully meet PLOS ONE’s publication criteria as it currently stands. Therefore, we invite you to submit a revised version of the manuscript that addresses the points raised during the review process.

We had trouble finding a second reviewer, so please respond carefully to the suggestions of the very comprehensive review that we were fortunately able to secure.

We look forward to receiving your revised manuscript.

Kind regards,

Christopher M. Danforth

Academic Editor

PLOS ONE

Journal Requirements:

Reviewers' comments:

Reviewer's Responses to Questions

**Comments to the Author**

1. Is the manuscript technically sound, and do the data support the conclusions?

Reviewer #1: Yes

2. Has the statistical analysis been performed appropriately and rigorously? 

Reviewer #1: I Don't Know

3. Have the authors made all data underlying the findings in their manuscript fully available?

Reviewer #1: Yes

4. Is the manuscript presented in an intelligible fashion and written in standard English?

Reviewer #1: Yes

5. Review Comments to the Author

Reviewer #1: Summary:

The authors introduce a Python module called PyPlutchik, which is built on top of popular plotting library MatPlotLib and provides an array of functions for visual representation of emotions in a circumplex, inspired by the infographic design by Robert Plutchik often cited in textual affect research. The authors provide ample evidence of the need within the Natural Language Processing community for a code implementation of this historically important graphical representation of affect, and include additional features to their module which show promise for more advanced and novel incarnations of the Plutchik wheel of emotions. In particular, the authors highlight the spatial relationships of the circumplex as integral to representation of semantic adjacencies and dichotomies; expand upon the notion of complex multi-faceted emotional affect, which greatly improves representation of nuance over singular categorizations for large corpora; and provide users with a range of options from simplistic representation to much more complex representations and small multiples, indicating that the authors have anticipated a range of research use cases. The authors demonstrate a steadfast commitment to the legacy of the original Plutchik Wheel (e.g. in choice of color and spatial orientation), but could perhaps benefit from a deeper acknowledgement of the critiques to the circumplex model in the broader research community since its inception. The graphics generated by the module demonstrate a careful attention to detail and commitment to visual clarity by the authors. Overall, the paper excels in its presentation and substance of scientific contribution, but there are a few necessary areas for revision that should be addressed before publication, particularly with the accompanying code and the implementation of dyadic visualization; and some minor suggestions that will likely be addressed during proofreading or are simply recommendations to possibly benefit from rewording for clarity.

Necessary areas for revision:

The introduction should include more context on the circumplex model of emotion cited in [52], specifically introducing the phrasing of a ‘circumplex’ to the paper rather than leaving that verbage in the bibliography. “Emotion circumplex” is a very common bigram in this area of research.

p.3: Citing the number of Google Scholar results is inappropriate for a scholarly journal, as word/phrase frequency is not indicative of importance on its own, and documents included on Google scholar extend beyond peer-reviewed scholarly research. This reviewer is in agreement that this tool is a meaningful and useful contribution to the scientific community, and the specific cited papers’ usage of alternative plots in section 2 is a convincing display of need, but the mention of the search method for finding those papers is unnecessary and the total paper count is an unreliable measure.

Github repository should include:

dependencies.txt

Example code shown on p.7-8

Some descriptions of implementation detailed in the paper should be removed from the paper and instead included in the module documentation, giving priority to discussion of the visual representation and affect model, rather than minutiae of implementation (this will also help to ‘future-proof’ work, as it is easier to update a Readme in a repository when changes arise in dependencies, than to update the code a published paper)

Document line 269 (page 7 code line 2):

from matplotlib.pyplot import plt

should be import matplotlib.pyplot as plt

this is a discrepancy between Python 2 and Python 3 wherein Python 3 no longer supports implicit imports

code line 3: ‘nrow’/‘ncol’ should be pluralized (‘nrows’/‘ncols’) per matplotlib documentation here: https://matplotlib.org/stable/api/_as_gen/matplotlib.pyplot.subplots.html

It should be noted that other NLP methods such as the “valence/arousal” measure do not align precisely with the Plutchik categories. It would appear, then, that using NLP libraries which categorize emotions based on the valence/arousal axes would be visually skewed by PyPlutchik not because of their content, but because of the measure used. This is a critical point to note in the paper and address explicitly.

Fig 13 / SemEval discussion: Please include code for these plots in the GitHub repository. In particular, the data pipeline from SemEval to the dyad plots is unclear, and makes it difficult to answer this reviewer’s question: Why are all the dyad values so negligibly small? It seems increasingly challenging to evaluate minute differences in affect between Plutchik plots when the maximum value, for example in Fig. 13 (v), is 0.03 and the corresponding plot area takes up only a few pixels. This appears to demonstrate either an issue in the data processing, a need to reconsider scaling when all values are miniscule (just as line plot axes are adjusted in range to fit the data shown, perhaps values should be normalized by the maximum value for all categories prior to plotting), or some combination of both.

Suggestions:

It seems that “module” would be a more appropriate term than “library” for all instances of “PyPlutchik Library” in the abstract and paper: https://dev.to/hamza/framework-vs-library-vs-package-vs-module-the-debate-3jpp

line 49: “This score can be interpreted as a binary flag that represents if emotion i was detected or not” - if i is in a range between 0 and 1, this is not a binary flag. Binary flags are booleans: 0 OR 1, not a float between 0 and 1. This language should be changed such that i corresponds to 8 emotions and Σ ia,b,c (where a,b,c, correspond to the 3 levels of intensity) is less than or equal to one. From the code, it does not appear true that “all emotions in a branch must sum to 1”, only that their sum must be less than or equal to 1.

Line 62: “for instance, it is respected also in user interfaces displaying Plutchik’s emotions” please cite

Line 91: “the exploitation of relationships among different artworks” - the phrasing here is confusing. By exploitation do you mean exploration? What is meant by ‘artworks’?

Line 93: Is it true that ManyEyes supported interactive visualization? This reviewer had not personally used the tool before its discontinuation, but this article specifically mentions lack of interactivity support: https://boostlabs.com/blog/ibms-many-eyes-online-data-visualization-tool/

General question for section 2: Is the inclusion of ManyEyes relevant, given its discontinuation? What about other existing tools for visualization such as Tableau, Plotly, MatPlotLib, Bokeh, Shiny, D3, or DataWrapper, to name a few? Tools currently in use seem like more relevant STAR benchmarks for discussion.

Section 2: Table 1 would benefit from accompanying representation of the original Plutchik emotion circumplex, e.g. https://www.researchgate.net/profile/Gerardo-Maupome/publication/258313558/figure/fig1/AS:297350959517696@1447905401049/Plutchiks-wheel-of-emotions-with-basic-emotions-and-derivative-emotions-and-cone-like.png

Line 141: Simplistic visual representation of variant dyad types would aid reading comprehension

There is no mention of cultural variation of emotion within the paper; however, the introduction of dyads does in part mitigate this oversight from a technical standpoint, by providing for a more complex description of emotion. Nonetheless, some acknowledgement/discussion of cultural differences in emotion should be included for context.

Please review citations carefully. A few examples:

Citation 22: “a ective” → affective

Citation 5: “Nrclex” → “NRClex” (please include URL)

Citation 24: “Imdb” → “IMDB” and “kaggle” → “Kaggle”

29: “fmcg” capitalize

“facebook”, “twitter”, “Italian” - brands & languages/countries should always be title cased

66: “Manyeyes” → “ManyEyes”

etc.

6. PLOS authors have the option to publish the peer review history of their article (what does this mean?). If published, this will include your full peer review and any attached files.

Reviewer #1: No

---

## [Author Response · Author response to Decision Letter 0]

29 Jul 2021

Turin, 26/07/2021

We are sincerely grateful to the reviewer for such a thorough and spot-on review. Their observations on our work have been deeply appreciated, as they pointed out many mistakes we needed to fix that survived our passes, and gave us a few useful insights that we discussed among us during the revision process. After this round of review, we feel that the overall quality of the manuscript has significantly improved. We also thank the reviewer for the careful checking of the code and for their suggestions. In the following pages we address every remark made by the reviewer, hoping that every answer is on point and clear enough. Of course we are always available for any further discussion or clarification.

With our best regards,

Alfonso Semeraro

Salvatore Vilella

Giancarlo Ruffo

Necessary areas for revision:

-The introduction should include more context on the circumplex model of emotion cited in [52], specifically introducing the phrasing of a ‘circumplex’ to the paper rather than leaving that verbage in the bibliography. “Emotion circumplex” is a very common bigram in this area of research.

We modified the introduction following the reviewer’s advice: we gave more space to Russel’s circumplex model, as well as to the circular representation of emotions (emotions circumplex) which is typical of many emotional models. Specifically, we added the following sentences:

“One of the most famous dimensional models is Russel's circumplex model of emotions [54]. Russel's model posits that affect states are defined by dimensions that are not independent: emotions can be represented as points in a space whose main orthogonal axes represent their degree of arousal and of pleasure/displeasure. According to Russel, the full spectrum of emotions can be meaningfully placed along the resulting circumference. Russel is not the first scholar to use a circular layout to represent emotions (for instance, it was already used in [33]): indeed, this kind of circumplex representation became very popular over time, since it is suitable to spatially represent emotions on a continuous space.“ 

We also highlighted that Plutchik’s model adopts a circumplex-like representation: even though it is not a continuous model, it makes use of a circular disposition of emotions.

-p.3: Citing the number of Google Scholar results is inappropriate for a scholarly journal, as word/phrase frequency is not indicative of importance on its own, and documents included on Google scholar extend beyond peer-reviewed scholarly research. This reviewer is in agreement that this tool is a meaningful and useful contribution to the scientific community, and the specific cited papers’ usage of alternative plots in section 2 is a convincing display of need, but the mention of the search method for finding those papers is unnecessary and the total paper count is an unreliable measure.

Following the reviewer’s suggestion, we removed “As of today, the query "Plutchik wheel" produces 3480 results on Google Scholar, of which 1620 publications have been submitted after 2017.” from the introduction. We also replaced

“the first 100 publications after 2017 retrieved by the aforementioned query, we notice that 25 over 100 papers” 

in Section 2 (line 188) with the less specific

“the related literature, we notice that many papers”.

-Github repository should include:

dependencies.txt

Example code shown on p.7-8

Some descriptions of implementation detailed in the paper should be removed from the paper and instead included in the module documentation, giving priority to discussion of the visual representation and affect model, rather than minutiae of implementation (this will also help to ‘future-proof’ work, as it is easier to update a Readme in a repository when changes arise in dependencies, than to update the code a published paper)

We added the requirements.txt file to GitHub repository, as requested. We also wrote a README.md, where we explained the main features of our module, and we displayed code examples and the corresponding plots.

We agree with the reviewer that most of the implementation details can be hidden from the paper, and moved elsewhere, as they are susceptible to changes in the future.

We removed the code listing and all the references to any parameter, which are now available in the code documentation, leaving in the paper only a qualitative description of the main features of the module. Documentation is available at

https://github.com/alfonsosemeraro/pyplutchik/blob/master/Documentation.md

We did not remove the references to the Python dict format as an input for the module, and we also kept Fig. 2, Fig. 3 and Fig. 6 (now Fig. 4, Fig. 5 and Fig. 8), including their small code listing that show the dict as the input data. We believe that part to be robust to changes in future implementations. It also advocates for the simplicity of the tool, as it features a familiar data structure and “how straightforward it is to plug a dict into the module to obtain the visualisation”, as we claim on line 291.

-Document line 269 (page 7 code line 2):

from matplotlib.pyplot import plt

should be import matplotlib.pyplot as plt

this is a discrepancy between Python 2 and Python 3 wherein Python 3 no longer supports implicit imports

code line 3: ‘nrow’/‘ncol’ should be pluralized (‘nrows’/‘ncols’) per matplotlib documentation here: https://matplotlib.org/stable/api/_as_gen/matplotlib.pyplot.subplots.html

We are really grateful to the reviewer for such a thorough review also of the code snippets, as in this case this one was definitely wrong. We fixed it into the documentation, and removed it from the paper, as per the previous reviewer’s observation. 

-It should be noted that other NLP methods such as the “valence/arousal” measure do not align precisely with the Plutchik categories. It would appear, then, that using NLP libraries which categorize emotions based on the valence/arousal axes would be visually skewed by PyPlutchik not because of their content, but because of the measure used. This is a critical point to note in the paper and address explicitly.

This is indeed a very important point to address. We designed PyPlutchik to work only with data annotated following Plutchik’s scheme - we never thought about the possibility to use it with other data. We addressed this in the conclusions, by adding the following sentences:

“It should be noted that the PyPlutchik module is meant to work only with data that was classified - manually or automatically - according to the Plutchik's model of emotions. In general, it is never advisable to represent data on a different scheme from the one used for its classification. Most of the models are radically different from each other, both in the underlying assumptions as well as in the categories scheme; even when they are somewhat similar, there might be key differences that should be kept into account. As for the Plutchik model in particular, to the best of our knowledge it does not exist a one-to-one correspondence with any other discrete emotional model, as well as no methods to go from a continuous emotional space to a discrete one. For these reasons, PyPlutchik should be used only with data annotated following the Plutchik's scheme of annotation.”

-Fig 13 / SemEval discussion: Please include code for these plots in the GitHub repository. In particular, the data pipeline from SemEval to the dyad plots is unclear, and makes it difficult to answer this reviewer’s question: Why are all the dyad values so negligibly small? It seems increasingly challenging to evaluate minute differences in affect between Plutchik plots when the maximum value, for example in Fig. 13 (v), is 0.03 and the corresponding plot area takes up only a few pixels. This appears to demonstrate either an issue in the data processing, a need to reconsider scaling when all values are miniscule (just as line plot axes are adjusted in range to fit the data shown, perhaps values should be normalized by the maximum value for all categories prior to plotting), or some combination of both.

 The pipeline from the SemEval data to the dyad plots in Fig. 12 and 13 (which now are Fig. 14 and 15) is pretty straightforward: we flagged a dyad to be present in one row of the dataset if both the two basic emotions that compose such a dyad are flagged as present. The reason for such small scores relies on the data itself. 

Our guess is that this corpus was annotated considering the 8 basic emotions only, and no dyads, as they did not appear in the data. This makes the dyads more rare, because they were not specifically intended by the annotators. We experienced this effect in several projects we worked on, where we annotated emotions in texts: when we did not consider dyads in the annotation interface, then dyads happened to be rare in the annotated data. When dyads were encompassed by the annotation interface, annotators used to click on them more often.

As per the visualization, a parameter for rescaling the petal length on a given maximum is actually present in our module: it is the “normalize” parameter. We used it for both plots, but setting for all subplots the same maximum value (0.65) for consistency reasons: the high scores of the leftmost subplot made the other scores in the other subplots very small, in comparison. 

We now rescaled the dyads subplots with different scales (0.4 and 0.3). Due to the loss of consistency, Fig. 14 and Fig. 15 are not intended to be read for a comparison between subplots, but rather for a better comparison of petals within the same plot. We acknowledge that in most of the subplots a single outlier stands clear above all the other petals, despite our efforts to rescale the figure. However, we believe that the outstanding difference between dyads with high scores and dyads with close-to-zero scores must be preserved and highlighted by the visualization, as it is exactly the message this representation is designed to convey. For instance, petals in Fig. 14 (iv) are barely readable, but we kept the same scale of Fig. 14 (iii), as it serves as a visual comparison between dyads very frequent in the corpus (like hope) and dyads barely detected.

We added the following disclaimer in line 461:

“For this reason we resized the petal lengths in Fig. 14 (ii - v) and (vii - x) with a different scale than (i) and (v), and the same applies for Fig. 15. For instance, Trust petal in Fig. 14 (i) is almost to the maximum length, with a score equal to 0.63; Hope petal in (iii) is also almost at the maximum length, but with a score equal to 0.37. Both figures are therefore designed for a visual comparison of the petals within the same subplot, but not between two different subplots, due to the lack of consistency in the scale. However”...

Suggestions:

-It seems that “module” would be a more appropriate term than “library” for all instances of “PyPlutchik Library” in the abstract and paper: https://dev.to/hamza/framework-vs-library-vs-package-vs-module-the-debate-3jpp

We replaced every occurrence of “library” with “module”, when talking about PyPlutchik. We kept the definition of “library” for matplotlib and NRCLex, whose software suite is remarkably bigger than ours.

-line 49: “This score can be interpreted as a binary flag that represents if emotion i was detected or not” - if i is in a range between 0 and 1, this is not a binary flag. Binary flags are booleans: 0 OR 1, not a float between 0 and 1. This language should be changed such that i corresponds to 8 emotions and Σ ia,b,c (where a,b,c, correspond to the 3 levels of intensity) is less than or equal to one. From the code, it does not appear true that “all emotions in a branch must sum to 1”, only that their sum must be less than or equal to 1.

The reviewer is right, the actual statement is incorrect. 0 and 1 can be considered binary flags only for a binary annotation of a single text, i.e. if that emotion has been detected or not in a text. When talking about average amounts of an emotion in a corpus, the score is indeed not binary anymore. We simply removed that consideration (line 49), as we think it can only contribute to confusion.

Also, we replaced 

“all the scores of emotions belonging to the same branch must sum to 1” 

in line 71 with 

“the sum of all the scores of the emotions belonging to the same branch must be less than or equal to 1”

-Line 62: “for instance, it is respected also in user interfaces displaying Plutchik’s emotions” please cite

 When we wrote this part, we had in mind two interfaces we used in two projects we worked on, where we annotated Plutchik’s emotions in tweets. The first interface appears like a series of colored buttons, with the classic Plutchik’s color-code. The second interface looks like a clickable Plutchik’s flower. Unfortunately, we realized that none of them is currently published.

We then removed the sentence in line 62, as we are not aware of further annotation interfaces that reproduce such a graphic detail.

-Line 91: “the exploitation of relationships among different artworks” - the phrasing here is confusing. By exploitation do you mean exploration? What is meant by ‘artworks’?

We clarified this sentence by changing it to: 

“ManyEyes was designed as a web based community where users (mainly data analysts and visulisation designers) could upload their data to establish conversations with other users.”

-Line 93: Is it true that ManyEyes supported interactive visualization? This reviewer had not personally used the tool before its discontinuation, but this article specifically mentions lack of interactivity support: https://boostlabs.com/blog/ibms-many-eyes-online-data-visualization-tool/

 As explained in this video: https://www.youtube.com/watch?v=aAYDBZt7Xk0 ManyEyes allowed interactivity to some extent: it was possible to highlight parts of the figures, to scroll on maps, to refine the visualization. However, the link reported by the reviewer clearly states that interactive visualizations with ManyEyes were not possible. 

We removed such a controversial comment, as we believe it would not add any relevant contribution to the discussion of the current paper.

-General question for section 2: Is the inclusion of ManyEyes relevant, given its discontinuation? What about other existing tools for visualization such as Tableau, Plotly, MatPlotLib, Bokeh, Shiny, D3, or DataWrapper, to name a few? Tools currently in use seem like more relevant STAR benchmarks for discussion.

We believe it is not strictly necessary to include ManyEyes: at this point we are not so sure how relevant it is in the community since when it has been included in IBM Watson. We can leave it there for good measure, and we follow the suggestion of the reviewer by mentioning many other tools that are probably highly used by scientists in this research area. We added the following sentence:

“nowadays, it exists a good number of software suites - such as Tableau, Microsoft PowerBI or Datawrapper, just to mention a few - that give the user a chance to create very interesting, eye-catching and often complex visualizations. They all adopt a graphical user interface, with all the pros and cons that usually come with it: an intuitive and fast way to realise the majority of the most common layouts, but likely less flexibility when it comes to create a more personalised visualisation. On the other hand, programming libraries and modules - such as Python's Matplotlib [25], Plotly [79] and Bokeh [77], or D3.js [78] in Javascript allow the users to create freely their own visualisations, though with a much steeper learning curve for those who are not familiar with these technologies.”

-Section 2: Table 1 would benefit from accompanying representation of the original Plutchik emotion circumplex, e.g. https://www.researchgate.net/profile/Gerardo-Maupome/publication/258313558/figure/fig1/AS:297350959517696@1447905401049/Plutchiks-wheel-of-emotions-with-basic-emotions-and-derivative-emotions-and-cone-like.png

 We added what is now Fig. 2, a reproduction of the figure suggested by the reviewer, which we could not claim authorship for.

-Line 141: Simplistic visual representation of variant dyad types would aid reading comprehension

We added what is now Fig. 3. We made up this picture in the same fashion of the many diagrams that display Plutchik’s dyads available online. Combination of emotions one, two, three or four petals away into primary, secondary, tertiary and opposite dyads are explained by a small visual example.

-There is no mention of cultural variation of emotion within the paper; however, the introduction of dyads does in part mitigate this oversight from a technical standpoint, by providing for a more complex description of emotion. Nonetheless, some acknowledgement/discussion of cultural differences in emotion should be included for context.

We sincerely thank the reviewer for this comment: indeed this is a very important factor that we forgot to mention. We added in the introduction the following sentences to discuss how cultural variation can impact emotion modeling in general:

“The cultural variation of emotions has also been studied. This is a crucial factor to take into account when classifying emotions: intuitively, the categories themselves can radically change depending on the cultural background [82]. This is also valid with respect to how the annotator perceives the emotions in a text and, if the annotated data is then used to train a classifier, it can introduce a bias in the model. The nomenclature of emotions, their meanings and the relations of words to emotion concepts depend on the social frameworks in which we are born and raised: therefore, cultural variation can significantly impact this kind of analysis. In this regard, in [81] the authors estimate emotion semantics across a sample of almost 2500 spoken languages, finding high variability in the meaning of emotion terms, but also evidence of a universal structure in the categorization of emotions. Following different methodological approaches, similar results were previously obtained in [83]”

and we also stated that, as the reviewer correctly points out, the presence of dyads in Plutchik’s model partially makes up for this matter. At the beginning of Section 4, we added the sentences:

“This feature dramatically enriches the spectrum of emotions, beyond the primary ones; enriching the spectrum increases the complexity of the models, also contributing to mitigate, at least partially, the phenomenon of cultural variation discussed in Section 1.”

-Please review citations carefully. A few examples:

Citation 22: “a ective” → affective

Citation 5: “Nrclex” → “NRClex” (please include URL)

Citation 24: “Imdb” → “IMDB” and “kaggle” → “Kaggle”

29: “fmcg” capitalize

“facebook”, “twitter”, “Italian” - brands & languages/countries should always be title cased

66: “Manyeyes” → “ManyEyes”

etc.

We fixed all the typos and errors reported by the reviewer.

---

## [Editor Report · Decision Letter 1]

9 Aug 2021

PyPlutchik: visualising and comparing emotion-annotated corpora

PONE-D-21-18896R1

Dear Dr. Vilella,

We’re pleased to inform you that your manuscript has been judged scientifically suitable for publication and will be formally accepted for publication once it meets all outstanding technical requirements.

Kind regards,

Christopher M. Danforth

Academic Editor

PLOS ONE
---

## [Editor Report · Acceptance letter]

23 Aug 2021

PONE-D-21-18896R1 

PyPlutchik: visualising and comparing emotion-annotated corpora 

Dear Dr. Vilella:

I'm pleased to inform you that your manuscript has been deemed suitable for publication in PLOS ONE. Congratulations! Your manuscript is now with our production department. 

Kind regards, 

on behalf of

Dr. Christopher M. Danforth 

Academic Editor

PLOS ONE